# Role of Seipin in Human Diseases and Experimental Animal Models

**DOI:** 10.3390/biom12060840

**Published:** 2022-06-17

**Authors:** Yuying Li, Xinmin Yang, Linrui Peng, Qing Xia, Yuwei Zhang, Wei Huang, Tingting Liu, Da Jia

**Affiliations:** 1West China Pancreatitis Centre, Centre for Integrated Traditional Chinese Medicine and Western Medicine, West China-Liverpool Biomedical Research Centre, West China Hospital, Sichuan University, Chengdu 610041, China; 2020224020178@stu.scu.edu.cn (Y.L.); yangxinmin@wchscu.cn (X.Y.); xiaqing@medmail.com.cn (Q.X.); 2Department of Endocrinology and Metabolism, West China Hospital of Sichuan University, Chengdu 610041, China; linruipeng@stu.scu.edu.cn (L.P.); doczhangyw@scu.edu.cn (Y.Z.); 3Institutes for Systems Genetics & Immunology and Inflammation, Frontiers Science Center for Disease-Related Molecular Network, West China Hospital, Sichuan University, Chengdu 610041, China; 4Key Laboratory of Birth Defects and Related Diseases of Women and Children, Department of Paediatrics, West China Second University Hospital, State Key Laboratory of Biotherapy and Collaborative Innovation Center of Biotherapy, Sichuan University, Chengdu 610041, China; jiada@scu.edu.cn

**Keywords:** seipin, *BSCL2*, CGL2, PELD, *BSCL2*-associated motor neuron diseases

## Abstract

Seipin, a protein encoded by the Berardinelli-Seip congenital lipodystrophy type 2 (*BSCL2*) gene, is famous for its key role in the biogenesis of lipid droplets and type 2 congenital generalised lipodystrophy (CGL2). *BSCL2* gene mutations result in genetic diseases including CGL2, progressive encephalopathy with or without lipodystrophy (also called Celia’s encephalopathy), and *BSCL2*-associated motor neuron diseases. Abnormal expression of seipin has also been found in hepatic steatosis, neurodegenerative diseases, glioblastoma stroke, cardiac hypertrophy, and other diseases. In the current study, we comprehensively summarise phenotypes, underlying mechanisms, and treatment of human diseases caused by *BSCL2* gene mutations, paralleled by animal studies including systemic or specific *Bscl2* gene knockout, or *Bscl2* gene overexpression. In various animal models representing diseases that are not related to *Bscl2* mutations, differential expression patterns and functional roles of seipin are also described. Furthermore, we highlight the potential therapeutic approaches by targeting seipin or its upstream and downstream signalling pathways. Taken together, restoring adipose tissue function and targeting seipin-related pathways are effective strategies for CGL2 treatment. Meanwhile, seipin-related pathways are also considered to have potential therapeutic value in diseases that are not caused by *BSCL2* gene mutations.

## 1. Introduction

Seipin, an evolutionarily conserved protein encoded by the Berardinelli-Seip congenital lipodystrophy type 2 (*BSCL2*) gene, is found to locate in the endoplasmic reticulum (ER) [1,2]. SEIPIN was initially discovered in type 2 congenital generalised lipodystrophy 2 (CGL2) patients, implying that it plays an important role in adipose tissue homeostasis [2]. Lipid droplets (LDs), the main organelles for storing triacylglycerols (TAGs) in adipose tissue, play a key role in lipid homeostasis and normal cell physiology [3]. Seipin belongs to an array of ER proteins including fat-storage-inducing transmembrane protein 2 (FIT2), Atlastin, receptor-expression-enhancing protein 1 (REEP1), etc., that are involved in LD biogenesis. Seipin deficiency can result in abnormalities of LDs in terms of number and morphology, which eventually lead to the loss of adipocyte tissue [4]. Therefore, the structure and function of seipin have been increasingly researched and summarised [5,6,7].

In recent years, numerous findings have been unravelled by studying the role of seipin in diverse diseases. Mutations of *BSCL2* damage different organs, such as the brain [8,9,10], mammary gland [11,12], pancreatic islets [13], and testis [14,15], which lead to the development of mutant seipin-induced genetic diseases including CGL2, progressive encephalopathy with or without lipodystrophy (PELD), also called Celia’s encephalopathy, and *BSCL2*-associated motor neuron diseases (also termed as “seipinopathies” by Ito and Suzuki). Both CGL2 and PELD are inherited as autosomal recessive, whereas *BSCL2*-associated motor neuron diseases are almost all autosomal dominant. Furthermore, the expression levels of seipin are found to be closely related with the progress of more common diseases not caused by *BSCL2* mutations, implying that seipin may have potential prognostic and therapeutic values in these diseases [16,17,18,19,20,21]. In this review, we integrate the findings of these papers, exploring the expression, regulation, and function of seipin, aiming to provide fundamental insights into its critical role in the pathogenesis of human diseases and experimental animal models.

## 2. Structure and Physiological Function of Seipin

The N-terminal and C-terminal of the ER-located seipin protein are exposed to the cytoplasm, and the two transmembrane helices wrapped in the ER lumen are connected by a large, highly conserved cavity ring [22]. Multiple seipin subunits form a cyclic protein complex [23], and the hydrophobic α-helices at the inner rim of the seipin protein complex are likely binding to the ER membrane [23,24]. While they remain elusive, existing studies have outlined the predominant functions of seipin as follows: (1) Seipin interacts with proteins involved in TAG synthesis to ensure normal adipogenesis [25,26]. Firstly, seipin deficiency significantly elevates glycerol-3-phosphate acyltransferase (GPAT) activity, resulting in retarded adipogenesis [25]. Secondly, seipin could bind to 1-acylglycerol-3-phosphate O-acyltransferase 2 (AGPAT2), phosphatidic acid (PA) phosphatase lipin 1, or both to regulate the PA metabolism [26]. The binding of seipin simultaneously to AGPAT2 and lipin 1 may facilitate PA clearance by supplying PA from AGPAT2 to lipin 1 [26]. Therefore, seipin deficiency may lead to abnormal PA accumulation [27,28,29]. On the nuclear envelope, the accumulation of PA traps and inhibits peroxisome proliferator activated receptor gamma (PPARγ) and thus blocks adipogenesis [30,31]. The lack of binding between seipin and AGPAT2 also contributes to reduced nuclear PPARγ accumulation, but the underlying mechanism requires further investigation [26]. On the ER, the accumulation of PA increases surface tension and decreases line tension of local ER, impeding unidirectional LD budding and normal ER-LD contact [32]. (2) The structural role of seipin in TAG and diacylglycerol (DAG) facilitates TAG phase separation, LD budding, and growth [33,34,35]. Seipin protein complex not only captures TAG and DAG inside its ring, but also provides TAG clusters for multiple nucleation sites [33,34]. Recently, the seipin protein complex has also been found to form a flexible cage-like structure, and the dynamic conformation changes in this structure contribute to the TAG phase separation, as well as LD growth and budding [35]. (3) Compelling evidence indicates that seipin plays a key role in the structure of ER-LD contact to promote the biogenesis or maintenance of LDs [4,36,37,38,39]. Seipin facilitates LD initiation via connecting nascent LDs to ER and stabilising ER-LD contact to restrict the lateral movement of LDs [4,38]. Then, seipin promotes LD growth by transferring the cargo of ER proteins and lipids into nascent LDs [38]. Thereafter, seipin further prevents the shrinkage of LDs during TAG flow from smaller to larger LDs, a process called ripening [35], through mobilising TAGs to small LDs [4,37]. Moreover, Fld1 (yeast seipin) cooperates with its partner Pex30 to promote LDs budding and pre-peroxisomal vesicle budding by stabilising ER domains [39]; the Fld1/Ldb16 complex establishes a diffusion barrier, which is necessary for LD surface tension and identity via stabilising ER-LD contacts to avoid phospholipid packing defects and abnormal LDs formation [40]. (4) Seipin promotes fat storage in adipose tissue by regulating calcium homeostasis and mitochondrial energy [41,42,43]. Seipin is enriched in ER–mitochondria contact sites nearby calcium regulators (sarco/ER calcium ATPase pump 2, inositol 3 phosphate receptor, and voltage-dependent anion channel), and interacts with them in a nutritionally regulated manner [43]. Therefore, seipin deficiency results in mitochondrial calcium import defects, leading to impairment of the Krebs cycle [42,43]. Since citrate and ATP are critical for lipogenesis and adipocyte properties, respectively, defects in citrate and ATP suppress fat storage [42,43]. (5) Seipin deficiency also compromises adipocyte properties via inducing ER stress in adipocyte tissue [43,44]. (6) Seipin deficiency accelerates lipolysis. This phenomenon is only observed during the early stage of seipin loss because there are rare TAGs left after this period [43,45,46]. Elevated adipose triglyceride lipase (ATGL) stability and expression, increased phosphorylation of hormone-sensitive lipase (HSL) and perilipin 1 (PLIN1) contribute to seipin deficiency-induced lipolysis acceleration [47,48]. Additionally, seipin has been found to participate in maintaining metabolic homeostasis via recruiting remodelling actin cytoskeleton of adipocytes [49], protecting mature adipocyte tissue [44,46], and restricting lipogenesis as well as LD accumulation in non-adipocyte tissue [50].

Physiologically, seipin has tissue-specific functions. In the adipose tissue, seipin serves as an adipocyte safeguard that maintains normal neutral lipid storage. In the liver, seipin is crucial for hepatic intracellular TAG accumulation [16]. Interestingly, despite adipose tissue and the liver storing neutral lipid in physiological conditions, seipin adipose-specific knockout (seipin-aKO) and liver-specific knockout (seipin-lKO) mice show distinctively different phenotypes. Seipin-aKO mice show overt lipodystrophy and hepatic steatosis [44], while seipin-lKO mice neither develop hepatic steatosis even under a high-fat diet, nor have worsened metabolic disorders on top of a seipin-aKO background [51,52]. Findings from seipin-aKO and seipin-lKO mice suggest that the normal TAG storage function of adipose tissue protects against liver steatosis. In the central nervous system, seipin is also widely expressed and plays a key role in intellectual development and motor neuron function [53]. It has been shown that seipin contributes to the deranged metabolism of phospholipids in the brain: seipin deficiency induces altered phospholipid components in the membrane, which may be associated with defective synaptic vesicle budding; seipin deficiency may also impact synaptic vesicle formation via affecting phospholipids, a main component of synaptic vesicles [54]. Furthermore, seipin has been linked to pre-peroxisomal vesicle biogenesis, which protects neurons from abnormal neuronal migration and differentiation, demyelination, inflammation, oxidative stress, and so on [55,56]. In the testis, seipin is essential for normal spermiogenesis by maintaining testicular phospholipid metabolic homeostasis [14].

## 3. Human Diseases Caused by Mutations of *BSCL2*

SEIPIN is widely expressed in human tissues, and the highest levels of this protein are found in the adipose tissue, brain, and testis [2,57]. Strikingly, the expression of SEIPIN in the brain is inversely correlated with age, but strongly and positively associated with anti-oxidative stress enzymes such as (superoxide dismutase 1 and 2 and PPARγ) [57]. Therefore, the mutations of *BSCL2* usually induce a variety of serious clinical consequences (Figure 1, Table 1 and Appendix A), including CGL2 (OMIM #269700; highest prevalence, estimated to be 0.1–5 persons per million [45,58]), PELD (OMIM #615924; only nine patients reported worldwide so far) [59], and *BSCL2*-associated motor neuron diseases (OMIM #619112 or OMIM #270685; secondary prevalence).

### 3.1. CGL2

CGL is a metabolic disease characterised by almost complete loss of adipose tissue and severe metabolic disorders, including early-onset diabetes mellitus, insulin resistance, hypertriglyceridaemia, and hepatic steatosis [58]. Variants of four different genes (*AGPAT2*, *BSCL2*, *CAV1*, and *PTRF*) induce CGL1, CGL2, CGL3, and CGL4, respectively. All subtypes of CGL lose metabolically active adipose tissue, but each of the subtypes has their unique features in addition to genetic testing (gold criteria) for differential diagnosis: (1) CGL1 patients usually suffer from focal lytic lesions in long bones after puberty; (2) CGL2 patients lose their mechanical adipose tissue except for metabolically active adipose tissue; (3) CGL3 patients are characterised by short stature, functional megaoesophagus, and hypocalcaemia; (4) CGL4 patients may progressively lose their adipose tissue during infancy, and face the challenge of congenital myopathy with high serum levels of creatine kinase [58]. CGL2 accounts for 50.3% of all four CGL distinct subtypes and has the most serious clinical consequence [58,72]. Up to now, there have been no reports of phenotypic differences among different mutation forms in CGL2 patients [58], but a few studies suggest that the severity of CGL2 is associated with sex, with females being more severely affected [73,74]. Since CGL2 patients have systemic multi-organ damage, we endeavour to characterise the phenotype of CGL2 patients from metabolic disorders and their complications, physical features, neurological symptoms, reproductive system symptoms, and others (Table 2).

Compared with other types of CGL, the loss of adipose tissue in patients with CGL2 is more serious [75,76]. In addition to the absence of metabolically active adipose tissue, a paucity of mechanical adipose tissues in palms, soles, eyes, scalp, and areas around joints is also noted in CGL2 patients [75]. Therefore, CGL2 patients suffer from earlier onset of diabetes mellitus, more severe insulin resistance, and a higher risk of premature death than other forms of CGL [61,62,77]. Recently, a study revealed that lipid peroxidation is increased in blood from CGL2 patients, which may be caused by an imbalance of redox homeostasis and increased mitochondrial DNA damage levels [78]. Cardiac complications and renal injury induced by metabolic disturbance are usually observed in CGL2 patients [58,79]. Cardiac complications in CGL2 patients are prominently characterised by concentric left ventricular hypertrophy, which is associated with diastolic dysfunction but preserved systolic function and ultimately leads to ventricular dysfunction and heart failure [48,80]. Additionally, cardiac complications in CGL2 patients have higher morbidity (42.9%) and poorer prognosis than other types of CGL, which is more likely to cause heart failure and death [58,61,81,82,83]. CGL is usually associated with proteinuria and renal injury [84]. Of 11 patients with CGL2, 4 developed nephropathy in a study from Turkey [63]. In another study, 2 out of 45 CGL2 patients died from renal failure [60]. Renal anomaly (absence of one kidney) and bilateral renal hypertrophy have also been reported [85,86], but these may be due to individual differences because only one case of each problem has been reported. In addition to hepatomegaly caused by hepatic steatosis, splenomegaly is also usually observed in CGL2 patients [85,87,88,89,90]. Further, acute pancreatitis has also been reported in a 4-month-old CGL2 infant who has concomitant extremely high serum triglyceride levels (31.9 mmol/L) [82]. High bone mineral density has also been noted in CGL2 patients, and loss of marrow adipose tissue, severe insulin resistance, and low adipokines levels may account for this phenomenon [64,91,92].

CGL2 patients often have many physical changes, mainly including acanthosis nigricans, hirsutism, muscular hypertrophy, and acromegaly [65,90,93]. Additionally, triangular facies [65,89,90], large ears [74,89,94], prognathism [60], and hernia [14,89,90] have all been reported.

Apart from the typical features of metabolic disturbances, CGL2 patients also have a greatly increased probability of developing mild to moderate mental disability [58,60,61,79,95]. Occasionally, symptoms of motor neuron impairment such as waddling gait [60], spastic gait [95], and developmental language disorders [93] have been reported.

CGL2 patients also suffer from impairment of the reproductive system. Effects on male patients are more severe, mainly manifested as enlarged external genitalia and infertility [14,86]. Teratozoospermia has been reported in a CGL2 male patient, with sperm defects including abnormal head morphology, ectopic accumulation of large LDs, and aggregation in dysfunctional bundles [14]. Female CGL2 patients often have precocious puberty [60], primary amenorrhoea [96], intrauterine growth restriction [60,89], polyhydramnios [60], oligomenorrhea [97], and polycystic ovary syndrome [63,97,98].

Some studies suggest that CGL2 patients are prone to inflammation, including upper respiratory tract infection [85,99], skin abscess [85], and urinary infections [100], while COVID-19 infection does not significantly worsen clinical outcomes [101]. Bone cysts [60,89], nubecula [87], higher total-body zinc clearance [102], and elevated urinary organic acid [99] associated with CGL2 patients have also attracted the attention of researchers. However, whether these manifestations are caused by seipin deficiency and potential underlying mechanisms need further study.

### 3.2. PELD

PELD is a fatal paediatric neurodegenerative disease manifested as severe progressive neurodegeneration and concomitant premature death caused by status epilepticus or pneumonia [59]. PELD is induced by a special nonsense mutation-c.985C > T in the *BSCL2* gene, and the mutation can be homozygous or in compound heterozygosity with c.507_511del, c.538G > T, or c.1004A > C [1,103]. Recently, Sánchez-Iglesias et al., have comprehensively summarised the diagnosis, treatment, and mechanism of PELD [59]. The mutation can be homozygous or in compound heterozygosity with other mutations

### 3.3. BSCL2-Associated Motor Neuron Diseases

Single heterozygous, compound heterozygous, or homozygous missense mutation (N88S or S90L) in *BSCL2* causes a broad spectrum of motor neuron diseases, not CGL2 [1,67]. These individuals do not suffer from fat metabolism disorder or abnormal distribution of body fat, but do suffer from many motor neuron diseases, such as Silver Syndrome (also known as spastic paraplegia type 17), distal hereditary motor neuropathy type V, and Charcot-Marie-Tooth type 2 [1,67], which are termed “seipinopathies” by Ito and Suzuki [104]. However, following the terminology used for diseases caused by lamin A/C (*LMNA*) gene mutations, all diseases caused by *BSCL2* mutations (CGL2, PELD, and *BSCL2*-associated motor neuron diseases) should be called “seipinopathies” [59,105]. Therefore, we call “seipinopathies termed by Ito” other than CGL2 and PELD as “*BSCL2*-associated motor neuron diseases” as per a dyadic nomenclature in this review. The main manifestations of patients with *BSCL2*-associated motor neuron diseases are distal limb muscle weakness and atrophy, hypertonia, abnormal reflex, gait abnormalities, and so on [67,71]. There is phenotypic heterogeneity observed in motor neuron damage caused by the different variant (heterozygous N88S or S90L mutation) of *BSCL2* [106,107], but the muscle magnetic resonance imaging pattern is consistent in that thenar eminence, soleus, and tibialis anterior are most frequently involved [71]. Two distinct simple heterozygous (c.566T > A, c.445C > G) missense *BSCL2* variants are associated with epileptic encephalopathy, which have been reported in three patients [105,108]. Furthermore, a single *BSCL2* variant including p.N88T [109], p.S141A [109], or p.R96H [110] may also result in one patient with motor neuron diseases, respectively. Details of phenotypes caused by these mutations still warrant further study.

### 3.4. Treatment for Human Diseases Caused by BSCL2 Mutations

The clinical treatment for diseases caused by *BSCL2* mutations is mainly directed towards symptoms [58], and we summarise studies that report specific drug therapy here (Table 3). In two CGL2 patients, it has been shown that pioglitazone improves lipodystrophy, insulin resistance, diabetes mellitus, and hypoleptinaemia [111,112]. In 30 CGL2 patients, leptin-replacement therapy improves metabolic disorders including diabetes mellitus, insulin resistance, hypertriglyceridaemia, fatty liver, and elevated plasma angiopoietin-like protein 3 [113,114,115,116,117,118]. Leptin-replacement therapy may be associated with normalising regular menses of female CGL2 patients. A 12-year-old female CGL2 patient who had not yet started her menses, did do so after receiving leptin. Another seven female patients between the ages of 15 and 40 years with lipodystrophy (5 patients with CGL1 and 2 patients with acquired generalised lipodystrophy) have commenced regular menses from primary amenorrhea or abnormal menstruation after this therapy [96]. Finally, in two PELD patients, leptin-replacement therapy delayed the neurological regression and allowed better seizure control [119,120].

## 4. Research Advances Obtained from Experimental Animal Models of *Bscl2* Mutations

### 4.1. Role of Seipin in CGL2

In 2011, Cui et al., established the first CGL2 animal model in mice, also named seipin knockout (SKO) mice, by *Bscl2* ablation through homologous recombination in embryonic stem cells [121]. The SKO mice recapitulate most of the metabolic phenotypes of human CGL2 patients, including significant loss of adipose tissue mass, glucose intolerance, liver steatosis, and hyperinsulinaemia [121]. However, in contrast to CGL2-associated hypertriglyceridaemia, SKO mice have hypotriglyceridaemia. This hypotriglyceridaemia may be due to increased clearance of TAG-rich lipoproteins and uptake of fatty acids by the liver, with reduced basal energy expenditure in mice [121,122]. Cardiomyopathy [123] and renal injury [124] induced by metabolic disorders in SKO mice are also noted. Consistent with CGL2 patients, the SKO mice also experience abnormalities in their nervous and reproductive systems. The neurological symptoms of SKO mice are mainly as follows: anxiety [8], depression [8,125], spatial cognitive deficits [9], and impairment of motor coordination [10]. For reproductive system changes in SKO mice, the males have both teratozoospermia and complete infertility [14,15], while all females have accelerated mammary gland ductal growth but delayed vaginal opening [11], insufficient milk production [12], and defective parturition [126]. The above phenotypes of this CGL2 animal model were further validated in mice by using another gene-editing technology (CRISPR-Cas9) to delete *Bscl2* in 2020 [127]. 

Since the expression pattern of seipin in the organs of rats is more similar to humans than in mice, Ebihara et al., established the first CGL2 rat model by using the method with ENU (N-ethyl-N-nitrosourea) mutagenesis for SKO in 2015 [128]. It has been found that the SKO rats are hypertriglyceridaemic, implying that it is a better model to represent blood lipid patterns in metabolic disorders observed in CGL2 patients [128]. The SKO rats also experience the impairment of memory abilities and infertility with azoospermia [128].

#### 4.1.1. Role of Seipin in Metabolic Disorders

Metabolic disorders are the predominant characteristics of SKO mice and attract numerous researchers investigating the underlying mechanisms and potential therapeutic targets (Table 4). Magré and Prieur have elegantly summarised the phenotypes and underlying mechanisms of metabolic disorders and the complications of CGL2 in mice in a recent review [45]. Therefore, rather than replicating their findings, this review complements this by adding some other findings that are not mentioned in their review.

In the review, Magré and Prieur described that seipin deficiency causes severe lipodystrophy by impairing adipogenesis, accelerating lipolysis, damaging LD homeostasis and other unclear mechanisms [45]. Here, we want to discuss whether the metabolic disturbance is caused by adipose tissue loss in SKO mice. On the one hand, the following evidence supports the conclusion that the metabolic disorders are caused by seipin deficiency induced adipocyte tissue loss: (1) aP2-driven seipin deletion induced glucose intolerance, insulin resistance, and liver steatosis in 6-month-old mice [44]; (2) mice are more susceptible to high-fat-diet-induced insulin resistance and hepatic steatosis because of acquired *Bscl2* deletion on mature adipose tissue [46]; (3) transplantation of normal adipose tissue ameliorates insulin resistance, dyslipidaemia, and severe hepatic steatosis in SKO mice [139]; (4) seipin deficiency results in obvious metabolic disturbance of liver or skeletal muscle in SKO mice in a non-cell autonomous way [51,52,140]. On the other hand, there are also some researchers who hold the opposite opinion: (1) there is no overt metabolic dysfunction in adiponectin-driven seipin deletion male mice, even following a high-fat diet challenge [46,136]; (2) adiponectin-driven seipin-deletion female mice fail to show severe metabolic disturbance, and only when facing the challenge of a high-fat diet at thermoneutrality do they develop moderate metabolic dysfunction [137].

Additionally, accumulating evidence also puts forward that seipin also participates in other physiological functions: (1) Seipin may play an important role in insulin synthesis and secretion via regulating the expression of PPARγ in pancreatic islets, and this function is affected by oestradiol [13]. (2) Seipin also plays a key role in plasma cholesterol metabolism together with low-density lipoprotein receptor (Ldlr). It has been found that an atherogenic diet induces much higher plasma cholesterol (6000 mg/dL) in seipin and *Ldlr* double knockout mice compared with SKO mice (300 mg/dL) or *Ldlr* knockout mice (1000 mg/dL) alone [133]. (3) Seipin contributes to normal cardiac function. It is well known that SKO mice develop hypertrophic cardiomyopathy with diastolic dysfunction [45]. Researchers have studied the underlying mechanisms of it in SKO mice and in cardiomyocyte-specific seipin knockout (seipin-cKO) mice. In SKO mice, hypertrophic cardiomyopathy is mainly caused by diabetes mellitus [45]. However, in seipin-cKO mice, dilated cardiomyopathy with systolic dysfunction is mainly induced by increasing ATGL expression, excessive fatty acid oxidation (FAO), as well as a drastic reduction in cardiac lipidome [144]. (4) Seipin plays an important role in vascular function. Wang et al., found that seipin deficiency leads to a reduction in perivascular adipose tissue mass and adipose-derived relaxing factors secretion, impairing vessel contractility and relaxation [142]. Bruder-Nascimento et al., proposed that hypoleptinaemia in SKO mice induces increased vascular adrenergic contractility and impairs endothelium-dependent relaxation via PPARγ-dependent mechanism increasing NADPH oxidase 1 expression and reactive oxygen species production [141,146]. Additionally, seipin deficiency induces hypertrophic vascular remodelling in which the underlying mechanism remain unclear [141].

#### 4.1.2. Role of Seipin in Neurodegenerative Phenotypes

Seipin deficiency can result in neurodegenerative phenotypes through a variety of mechanisms, which can be summarised as the following aspects (Figure 2 and Table 5): (1) Neuron-specific seipin knockout (seipin-nKO) induces neuron damage through inhibiting PPARγ and its downstream targets [8,9,10,20,125]. Studies in SKO, seipin-nKO, and seipin-aKO mice show that seipin deficiency in neurons (both SKO and seipin-nKO), but not adipose tissue, causes neurological injury by inhibiting PPARγ [8,9]. Since then, more and more studies have focused on how seipin deficiency induces neurodegeneration via inhibiting PPARγ. In hippocampal CA1 regions, PPARγ inhibition could selectively suppress α-amino-3-hydroxy-5-methyl-4-isoxazole propionic acid receptor (AMPAR) expression through decreasing the activity of extracellular signal-regulated kinase (ERK) and cAMP-responsive element binding protein (CREB), resulting in N-methyl-D-aspartate (NMDA) receptor-dependent long-term potentiation (LTP) and spatial memory impairment [9]. An in vitro study also finds knockdown of *Bscl2* gene expression impairs excitatory post-synaptic currents via reducing levels of surface AMPAR [147]. In the hippocampal dentate gyrus, PPARγ inhibition not only impairs the proliferation of stem cells by inhibiting ERK phosphorylation but also inhibits the differentiation of progenitor cells, which is caused by the reduction in neurogenin 1 (Neurog1) and neurogenic differentiation 1 (NeuroD1) expression via inhibiting Wnt3 signal [125]. In hippocampal neurons, PPARγ inhibition induces hyperphosphorylation and aggregation of tau phosphorylation at Thr^212^/Ser^214^ and Ser^202^/Thr^205^ through increasing protein kinase B–mammalian target of rapamycin (AKT-mTOR) signalling, which not only suppresses autophagy, but also elevates the glycogen synthase kinase-3β (GSK3β) activity, eventually leading to axonal atrophy [20,148]. In dopaminergic neurons, PPARγ inhibition elevates activated GSK3β-induced α-synuclein phosphorylation and neuroinflammation, and decreased PPARγ caused α-synuclein aggregation, resulting in dopaminergic cells loss [10]. (2) Metabolic dysfunction caused by SKO mice also contributes to its neurodegenerative phenotypes. Insulin resistance is responsible for neuron injury through increasing tau phosphorylation at Ser^396^ induced by the c-Jun N-terminal kinase (JNK) pathway [148], and low adipokine levels have been certified to be involved in the pathogenesis of neurovegetative disease because leptin plays a key role in memory performance via maintaining hippocampal function, and adiponectin takes part in various brain functions in physiology [149,150,151]. 

#### 4.1.3. Role of Seipin in Reproductive Phenotypes

In experimental animal models of *BSCL2* mutations, the potential effects on the reproductive system are proposed (Table 6).

Studies in SKO, germ-cell-specific seipin knockout (seipin-gKO), and seipin-aKO mice show that the absence of seipin protein in germ cells (SKO and seipin-gKO), but not in adipose tissue, leads to infertility and teratozoospermia [14]. The underlying reasons for seipin-gKO-induced phenotype could be concluded as follows: Lipid accumulation caused by altered phospholipid metabolism in seipin loss germ cells may contribute to the structural defects in spermiogenesis and severely abnormal sperm [14]. Seipin may play a key role in DNA protection because *Bscl2* is strongly induced after exposure to genotoxic agents in mouse embryonic stem (mES) cells (but not in primary liver cells) [152]. Seipin deficiency increases chromocenter fragmentation and defective chromatin condensation in sperm cells, impairing sperm quantity and motility [15]. Additionally, in mice without seipin in germ cells, defective mitochondrial activity and resultant impaired sperm motility also contributes to infertility [15].

According to previous studies, phenotypes involving the female reproductive system induced in SKO mice mainly include abnormal mammary gland development, insufficient milk production, and defective parturition. Female SKO mice have longer and more dilated mammary gland ducts at 5 weeks old [11], and the downregulation of oestrogen receptor β in mammary gland may be responsible for this phenomenon because it may have an antiproliferative function as it does in the uterus [11,153,154]. All adipocytes, LDs, and seipin itself have a role in milk production. White adipocytes transform into milk-secreting epithelial cells during pregnancy and lactation [155]; seipin-involved formation of mature LDs is one step of milk production by mammary epithelial cells [4,156]. Seipin is not only one component of the milk lipid globule membrane [157], but it also protects mammary gland alveolar epithelial cells from ER stress to avoid poor differentiation and/or apoptosis to guarantee normal milk secretion [12]. As for defective parturition in female SKO mice, myometrial hypertrophy caused by metabolic disturbance and elevated autophagy induced by uterine luminal epithelium seipin loss could both be expected to be responsible for this [126]. 

### 4.2. Role of Different BSCL2 Variants in Motor Neuron Diseases

*BSCL2*-associated motor neuron diseases damage upper motor neurons, lower motor neurons, and peripheral motor axons differently [104]. To date, there are three animal models for *BSCL2*-associated motor neuron diseases, including N88S mutant seipin transgenic (N88S seipin Tg) mice [158], N88S seipin Tg zebrafish [159], and N88S/S90L mutant seipin transgenic (N88S/S90L seipin Tg) mice [160]. Both the N88S seipin Tg mice and N88S/S90L seipin Tg mice reproduce the most symptomatic phenotype of patients with *BSCL2*-associated motor neuron diseases, including the progressive spastic motor deficit, reactive gliosis in the spinal cord, and neurogenic muscular atrophy [158,160]. The main phenotype of N88S seipin Tg zebrafish is a reduction in spontaneous free swimming of zebrafish [159].

The potential molecular mechanisms are summarised as follows: (1) Whether ER stress should be considered as a predominant mechanism is a subject of debate. Mutations of N88S and S90L in *Bscl2* inhibit the glycosylation of seipin, which leads to the misfolded and aggregated proteins and the excessive accumulation of misfolded proteins, resulting in ER stress in neurons [1,161,162]. N88S seipin causes ER stress in vitro, which is partially attributed to a reduction in TAG content [159]. In contrast, some researchers find that ER stress is not obvious in both N88S/S90L seipin Tg mice and N88S seipin Tg zebrafish [159,160]. Therefore, further work is required to ascertain the definitive role of ER stress in *BSCL2*-associated motor neuron diseases. (2) N88S/S90L seipin expression results in the formation of inclusion bodies in the cytoplasm. Compared with normal cells, N88S/S90L seipin induces the formation of inclusion bodies in the cytoplasm in different cell types, including motor neurons at a 25–30 percent level in vitro [161,162]. N88S seipin Tg mice and N88S/S90L seipin Tg mice also show that inclusion bodies are formed in the cytoplasm of neurons [158,160]. The formation of inclusion bodies may be an adaptive machinery against the accumulation of N88S/S90L seipin in ER and its associated ER stress [158]. However, there are only aggresomes, but not inclusion bodies, reported in patients [1]. Ito et al., found that inclusion bodies in experimental animals or cells are different from aggresomes in humans, because both their component proteins and formation locations are different [1,162,163]. (3) Seipin N88S expression affects the LD morphology and decreases the TAG content of nervous tissue. Many small and sometimes clustering LDs, similar to LDs caused by seipin deficiency, are induced by N88S/S90L seipin in vitro. The expression of seipin N88S leads to a reduction in TAG storage in both motor neuron cells and zebrafish [159,164]. Seipin maintains axonal function in long corticospinal neurons together with Reep1 through regulating LD formation and morphology [165]. Additionally, N88S/S90L seipin significantly activates the autophagy pathway in the central nervous system of mice, eventually leading to motor neuron death [160].

## 5. Research Advances Acquired from Diseases Associated with Abnormal Expression of Seipin

The role of seipin in diseases not caused by *BSCL2* mutations has attracted more and more attention in recent years. At present, the relationships between seipin and metabolic [16], neurodegenerative [10,17,18,19,20,166], and other diseases [21,167,168] have been reported. 

It is proposed that some diseases not caused by *BSCL2* mutations affect the levels of seipin expression, and the abnormal seipin expression would in turn affect disease progression (Table 7). In hepatic steatosis, seipin expression is downregulated, which promotes intracellular TAG storage via calcium-depletion-induced ER stress, and adeno-associated virus-mediated seipin overexpression in liver suppresses intracellular TAG accumulation via increasing cytosolic calcium [16]. In patients with Parkinson’s disease, the proteomic analysis of substantia nigra shows that expression of seipin is decreased [17]. Intriguingly, seipin expression levels are significantly increased in experimental Parkinson’s disease models both in vitro and in vivo, and overexpression of seipin aggravates ER stress via the Grp94/Bip-ATF4-CHOP signalling pathway [166]. Seipin deficiency facilitated Aβ_25–35/1–42_-induced neuroinflammation via the PPARγ-GSK3β-TNF-α/IL-6 pathway in astrocytes, resulting in neurodegenerative disorders eventually [20,169]. In glioblastoma patients and glioblastoma cell lines, the expression of *BSCL2* is increased [167]. In ischaemia/reperfusion-induced cerebral damage, the expression of seipin is inhibited by elevated miR-187-3p (a microRNA which can bind to the protein-coding sequence of seipin), and suppression of seipin expression increases neuronal apoptosis via deficient autophagic flux and ER stress [18,19]. Furthermore, SKO mice have exacerbated neurological disorder and enlarged infarct size that may be caused by increased blood–brain barrier permeability, amplified ER stress, and elevated glucose levels, as well as decreased leptin and adiponectin levels [170]. In a cardiac hypertrophy model in mice, seipin deficiency accelerates diastolic heart failure after transverse aortic constriction because of abnormal myocardial calcium handling and enhanced ER stress [21]. Recently, researchers have also found that the expression of seipin is inhibited in the myocardium of an intrauterine growth restriction sheep model [168]. 

## 6. Lessons Learnt from Experimental Animals: Seipin as Therapeutic Target

To date, the main treatments for diseases caused by *BSCL2* mutations are directed towards symptoms without specific and targeted therapeutic options [58,59], and yet, research on targeted therapy for PELD and *BSCL2*-associated motor neuron diseases is limited. Here, we summarise that seipin deficiency causes phenotypes via inducing adipose tissue loss or impairing the seipin signalling pathway, and two main strategies can be proposed to treat CGL2: (1) restoration of adipose tissue itself or its function to treat metabolic complications resulting from adipocyte seipin deficiency; (2) tackling seipin or seipin upstream or downstream targets to alleviate the effects of seipin deficiency in organs in which seipin works in a cell-autonomous way.

The restoration of adipose tissue function, which includes improving TAG storage capacity and maintaining adipokine levels, is the main principle to treat metabolic disorders and complications in CGL2 [171]. The potential modalities for restoring adipose tissue function can be concluded as follows: (1) normal adipose tissue transplantation; (2) PPARγ agonists; (3) exogenous replacement for adipokines; (4) inhibition of GPAT3 or ATGL; (5) restoration of seipin expression in adipose tissue. Details are as follows: (1) In SKO mice, metabolic disturbance and its complications, including insulin resistance, severe hypoleptinaemia, and severe hepatic steatosis, as well as renal injury, are abolished by adipose tissue transplantation [124,139]. (2) In SKO mice, pioglitazone ameliorates insulin resistance [122], liver steatosis [122], and hypertrophic cardiomyopathy [135]. In comparison with pioglitazone, rosiglitazone treatment not only rescues metabolic disorders in SKO mice [13,44], but also alleviates hyperlipidaemia and atherosclerosis caused by simultaneous absence of seipin and Ldlr in mice [133]. However, in SKO rats, both pioglitazone and rosiglitazone fail to rescue the fatty liver via rescuing PPARγ expression in the liver and adipocyte tissue. Therefore, whether the potential that fatty liver could be rescued by PPARγ agonists is species-dependent or related to the methodology in which SKO animals are generated remains unclear [145]. (3) In SKO mice, leptin-replacement therapy not only rescues impaired vascular function [141,146], but also ameliorates kidney injury secondary to metabolic disturbance [124]. In parallel, fibroblast growth factor 21, another adipokine, has also been used to improve the metabolic profile of white adipose tissue to enhance insulin sensitivity and increase plasma adiponectin levels in SKO mice [134]. (4) GPAT3 ablation significantly improves insulin sensitivity and liver steatosis in SKO mice through increasing adipocyte tissue mass [127]. Similar to GPAT3, adipose tissue mass is tightly correlated with ATGL expression in SKO mice [48], and heterozygous deletion of ATGL significantly ameliorates metabolic disorders and cardiac hypertrophy; complete ablation of ATGL even fully reverses seipin deficiency-induced lipodystrophy [48]. (5) In SKO mice, the recovery of seipin expression in adipocyte tissue improves metabolic disturbance [131].

In organs where seipin works in a cell-autonomous way, tackling seipin-related PPARγ pathways is the main principle to ameliorate disorders of these organs in CGL2. In SKO mice, rosiglitazone treatment ameliorates anxiety [8], depression [8,125], spatial cognitive impairment [9], and motor impairment [10], as well as Aβ_25-35/1-42_-induced Alzheimer’s disease [20]. In seipin-nKO male mice, either rosiglitazone or oestradiol ameliorate anxiety and depression via rescuing PPARγ [8]. Rosiglitazone or oestradiol also work in rescuing pancreatic function in male mice with heterozygous deletion of seipin [13]. The downstream targets of PPARγ also show great therapeutic potential: a mitogen-activated protein kinase kinase (MEK) inhibitor blocks the rosiglitazone-rescued depression and spatial memory in seipin-nKO mice [9,125]; a GSK3β inhibitor ameliorates α-synuclein phosphorylation, neuroinflammation and the hyperphosphorylation and aggregation of tau protein in seipin-nKO mice [10,148]; and a phosphoinositide 3-kinase inhibitor or mTOR inhibitor accelerates p-tau and tau protein clearance, relieving axonal atrophy in the hippocampal neurons of seipin-nKO mice [148]. 

In addition, some treatments that cannot be concluded by these two strategies also ameliorate CGL2 phenotypes. In SKO mice, the sodium-glucose cotransporter 2 inhibitor dapagliflozin successfully reduces the O-GlcNAcylated protein levels and blocks the development of hypertrophic cardiomyopathy [135]; JNK inhibitor improves hippocampal neuron damage through reducing hyperphosphorylated tau at Ser396 induced by the insulin resistance cascading JNK pathway [148]; miR-187-3p inhibitor ameliorates ischemia-induced cerebral damage significantly by rescuing seipin expression to improve autophagic flux and relieve ER stress [18,19]; and genistein treatment accelerates vaginal opening but increases mammary gland area in SKO mice [11].

## 7. Conclusions and Perspective

In the current review, the summarised evidence suggests that seipin plays an important role not only in rare diseases induced by *BSCL2* mutations, but also in more common diseases that are not related to *BSCL2* mutations. Limitations exist in studies focusing on the relationships between seipin and diseases: (1) studies on differential expression patterns of seipin in different diseases and the underlying mechanisms require exploration; (2) the precise mechanism of seipin in the brain and testis where it works in a cell-autonomous way is still unclear; and (3) there are rare seipin signalling-pathway-based treatment options for relevant diseases. Therefore, further studies to address these limitations are awaited. Moreover, similar to seipin, other ER-located proteins involved in LD biogenesis such as fat storage-inducing FIT2, Atlastin, REEP1 and so on are also closely associated with human diseases. For example, FIT2 has been certified to play a key role in lipodystrophy [172], diabetes mellitus [173,174], and hepatocellular carcinoma [175], as well as doxorubicin-mediated cardiotoxicity [176], and the underlying mechanisms are also incompletely clear. Therefore, it is of great significance to study the precious mechanism of these proteins in diseases, and studies focusing on seipin are excellent examples of them.

## Figures and Tables

**Figure 1 biomolecules-12-00840-f001:**
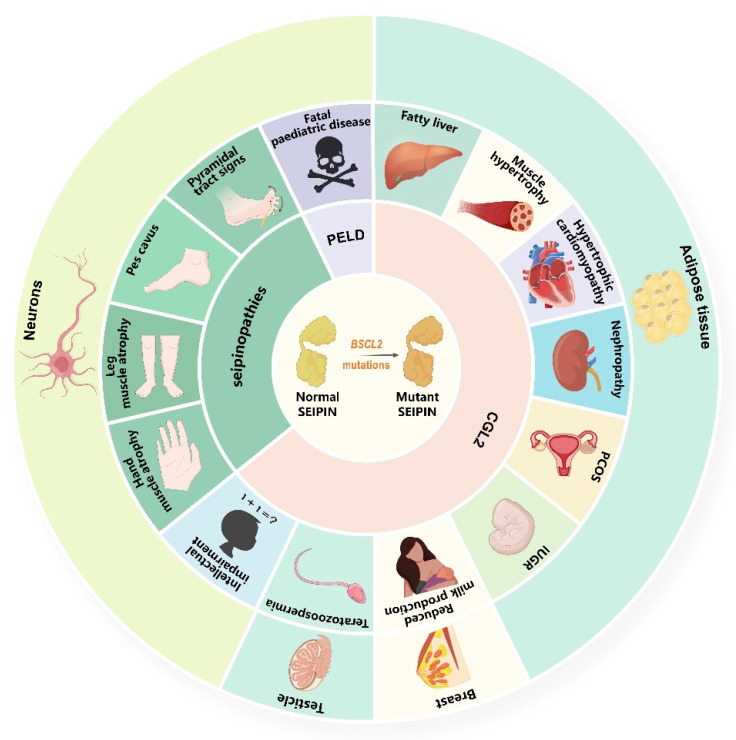
**Clinical features and organs damaged in diseases caused by *BSCL2* mutations.***BSCL2* mutations produce mutant seipin, and mutant seipin induces CGL2, PELD, and *BSCL2*-associated motor neuron diseases. CGL2 is characterised by metabolic disorders and its complications, which may be caused by adipose tissue loss. In addition, phenotypes resulting from seipin deficiency in breast, testicles, and neurons, including reduced milk production, teratozoospermia, and intellectual impairment, are also observed in CGL2. PELD is a fatal paediatric neurodegenerative disease. *BSCL2*-associated motor neuron diseases are predominantly damaging motor neurons, resulting in weakness and atrophy of distal muscle groups (hands and legs), foot deformity (pes cavus), and pyramidal tract signs. Abbreviations: *BSCL2*: Berardinelli-Seip congenital lipodystrophy type 2; CGL2: type 2 congenital generalised lipodystrophy; PELD: progressive encephalopathy with or without lipodystrophy; PCOS: polycystic ovary syndrome; IUGR: intrauterine growth restriction.

**Figure 2 biomolecules-12-00840-f002:**
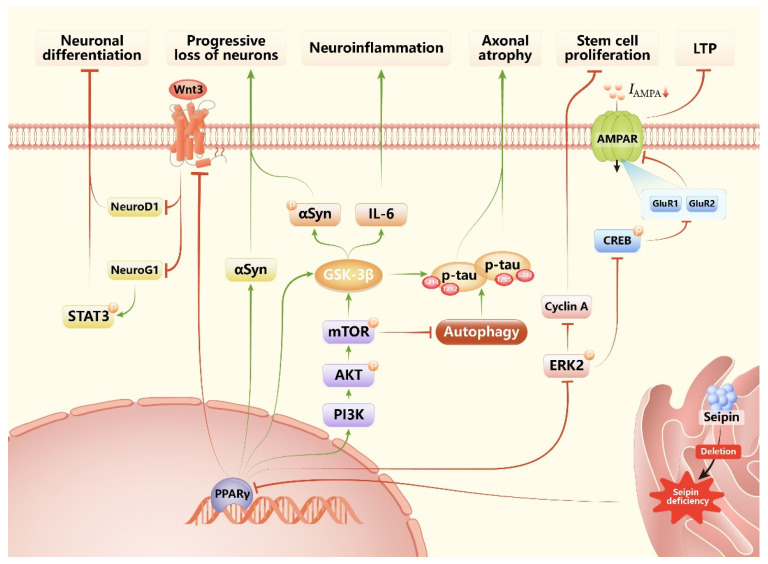
**Summary of the mechanisms in neurons leading to neurodegenerative phenotypes in SKO mice and in seipin-nKO mice.** Seipin deficiency suppresses PPARγ expression. Decreased PPARγ inhibits neuronal differentiation via the Wnt3 pathway, promotes the phosphorylation and aggregation of αsyn and tau protein and neuroinflammation through activating the PI3K/AKT/mTOR pathway and GSK-3β pathway, and impairs stem cell proliferation and the current of AMPAR, as well as LTP via inhibiting ERK pathways. Abbreviations: PPARγ: peroxisome proliferator activated receptor gamma; STAT3: signal transducer and activator of transcription 3; NeuroD1: neurogenic differentiation 1; Neurog1: neurogenin 1; αsyn: α-synuclein; PI3K: phosphatidylinositol 3-kinase; AKT: protein kinase B; mTOR: mammalian target of rapamycin; GSK3β: glycogen synthase kinase-3β; IL-6: interleukin-6; ERK2: extracellular signal-regulated kinase 2; CREB: cAMP-responsive element binding protein; GluR1: α-amino-3-hydroxy-5-methyl-4-isoxazolepropionic acid (AMPA) receptor subunits 1; GluR2: α-amino-3-hydroxy-5-methyl-4-isoxazolepropionic acid (AMPA) receptor subunits 2; AMPAR: α-amino-3-hydroxy-5-methyl-4-isoxazole propionic acid receptor; *I_AMPA_*: AMPA-induced current; LTP: long-term potentiation.

**Table 1 biomolecules-12-00840-t001:** Human disease cohorts (*n* ≥ 9) caused by *BSCL2* gene mutations.

Studies	Phenotype	OMIM/Inheritance	Patients or Families	Mutation Classification	Clinical Features and End-Organ Complications
Van Maldergem et al., 2002 [60]	CGL2	269700/AR	45 patients	Nonsense, missense, or frameshift	Metabolic disorders (diabetes mellitus, hyperlipidaemia) Changes in physical appearance (acanthosis nigricans, hirsutism, acromegaloid features, prognathism, vitamin D resistant rickets, facial dysmorphism, and macroglossia) Organ injury induced by metabolic disorders (hepatomegaly, liver cirrhosis, hypertrophic cardiomyopathy, angina pectoris, heart failure, renal failure, and organomegaly) Changes in reproductive system (precocious puberty, intrauterine growth restriction, polyhydramnios, and persistent Mullerian ducts) Neurodegenerative phenotypes (mild to moderate intellectual impairment, waddling gait) Others (bone cysts, multinodular goitre, and failure to thrive)
Agarwal et al., 2003 [61]	CGL2	269700/AR	17 patients	Frameshift or unknown	Metabolic disorders (diabetes mellitus, hypertriglyceridemia, hypoleptinaemia) Changes in physical appearance (acanthosis nigricans) Organ injury induced by metabolic disorders (cardiomyopathy) Neurodegenerative phenotypes (mental retardation)
Gomes et al., 2005 [62]	CGL2	269700/AR	22 patients	Frameshift	Metabolic disorders (diabetes mellitus, insulin resistance, hypertriglyceridemia, hypoleptinaemia)
Akinci et al., 2016 [63]	CGL2	269700/AR	11 patients	Nonsense or missense	Metabolic disorders (diabetes mellitus, hypertension) Changes in physical appearance (acanthosis nigricans, pseudo-acromegaly) Organ injury induced by metabolic disorders (elevated liver enzymes, cirrhosis, cardiomyopathy, proteinuria, diabetic neuropathy, on haemodialysis, diabetic foot ulcer, and retinopathy) Changes in reproductive system (polycystic ovary syndrome) Neurodegenerative phenotypes (mild mental retardation) Others (bone cysts)
Lima et al., 2018 [64]	CGL2	269700/AR	15 patients	NA	Metabolic disorders (diabetes mellitus, insulin resistance, hypertriglyceridemia, hypoleptinaemia) High bone mineral density in trabecular sites
Hsu et al., 2019 [65]	CGL2	269700/AR	16 patients	Nonsense, missense, frameshift, or unknown	Metabolic disorders (diabetes mellitus, insulin resistance, hypertriglyceridemia, hypoleptinaemia) Changes in physical appearance (acanthosis nigricans, muscular hypertrophy, and triangular face) Organ injury induced by metabolic disorders (hepatomegaly, elevated liver enzymes, cardiomyopathy) Neurodegenerative phenotypes (intellectual disability, loss of ambulation, and died prematurely)
Windpassinger et al., 2004 [1]	dHMN-V	619112/AD	9 families	Missense	Weakness and atrophy of the distal muscle groups (starting and predominating in the leg muscles)
	SS	270685/AD	7 families	Missense	Weakness and atrophy of the distal muscle groups (starting and predominating in the hand muscles)
Irobi et al., 2004 [66]	CMT-I	NA/AD	17 patients	Missense	Weakness and atrophy of the distal muscle groups (leg paralysis) Foot deformity (pes planus, pes cavus) Gait (loss of ambulation, spastic) Pyramidal tract damage (brisk reflexes in the lower limbs, Babinski sign)
	CMT-206	NA/AD	12 patients	Missense	Weakness and atrophy of the distal muscle groups Foot deformity (pes cavus, pes equinus) Gait (loss of ambulation, steppage, and abnormal)
Auer-Grumbach et al., 2005 [67]	dHMN-V	619112/AD	28 patients	Missense	Weakness and atrophy of small hand muscles Gait (abnormal) Preserved or slightly brisk tendon reflexes but normal muscle tone
	CMT	NA/AD	18 patients	Missense	Weakness and atrophy of the lower limb muscle groups preserved or slightly brisk tendon reflexes but normal muscle tone
	Subclinically affected	NA/AD	18 patients	Missense	Minor and nonspecific abnormalities
	SS	270685/AD	13 patients	Missense	Mild-to-severe symmetrical or unilateral amyotrophy of the small hand muscles A variable degree of spasticity of the lower limbs Pyramidal tract damage (brisk tendon reflexes, extensor plantar responses, and elevated muscle tone)
	HSP	NA/AD	9 patients	Missense	Spastic paraparesis in the lower limbs
	Asymptomatic	NA/AD	4 patients	Missense	Without any clinical abnormalities
van de Warrenburg et al., 2006 [68]	Overlapping SS-dHMN	NA/AD	12 patients	Missense	Weakness and atrophy of the distal muscle groups Foot deformity (pes cavus) Pyramidal tract damage (brisk reflexes in the lower limbs, Babinski sign)
Pennisi et al., 2012 [69]	dHMN-V	619112/AD	12 patients	Missense	Weakness and atrophy of the distal muscle groups Foot deformity (pes cavus) Gait (steppage, severely impaired)
Choi et al., 2013 [70]	CMT2	NA/AD	11 patients	Missense	Weakness and atrophy of the distal muscle groups (prominent involvement of thenar muscles) Foot deformity (pes cavus) Brisk biceps, ankle, or knee reflexes Gait (Spastic) Pyramidal tract damage (plantar response) Sensory loss (vibration sense was impaired more severely than pain sense)
Fernández-Eulate et al., 2020 [71]	dHMN-V	619112/AD	9 patients	Missense	Always upper and lower limb atrophy Foot deformity (pes cavus, hammer toes, Achilles retraction) Pyramidal tract damage (Babinski signs, brisk reflexes) Others (respiratory insufficiency)
	SS	270685/AD	5 patients	Missense	Weakness and atrophy of the distal muscle groups Foot deformity (pes cavus, hammer toes, Achilles retraction) Pyramidal tract damage (Babinski signs, brisk reflexes) Others (spasticity)
	dHMN-II	NA/AD	4 patients	Missense	Only lower limb atrophy Foot deformity (pes cavus, hammer toes, Achilles retraction) Pyramidal tract damage (Babinski signs, brisk reflexes)
	CMT2	NA/AD	3 patients	Missense	Always upper and lower limb atrophy Foot deformity (pes cavus, hammer toes, Achilles retraction) Pyramidal tract damage (Babinski signs, brisk reflexes) Sensory symptoms or sensorimotor neuropathy
	Asymptomatic	NA/AD	5 patients	Missense	One of them with foot deformity
Sánchez-Iglesias et al., 2021 [59]	PELD	615924/AR	9 patients	Nonsense	PELD is a fatal paediatric neurodegenerative disease which induces death through status epilepticus or pneumonia secondary to a progressive deterioration due to neurodegeneration.

Abbreviations: CGL2: type 2 congenital generalised lipodystrophy; AR: autosomal recessive; NA: not available; dHMN-V: distal hereditary motor neuropathy type V; AD: autosomal dominant; SS: silver spastic paraplegia syndrome; CMT: Charcot–Marie–Tooth disease; HSP: hereditary spastic paraparesis; PELD: Celia’s encephalopathy or progressive encephalopathy with or without lipodystrophy.

**Table 2 biomolecules-12-00840-t002:** Phenotypes of CGL2 patients.

**Metabolic disorders**	Diabetes mellitus, insulin resistance, mixed dyslipidaemia, hyperlipidaemia, hypertriglyceridaemia, hypercholesterolaemia, hypoleptinaemia, low high-density lipoprotein cholesterol, hypertension, hypertransaminasaemia
**Organ injury induced by metabolic disorders**	Hepatomegaly, fatty liver/hepatic steatosis, elevated liver enzymes, steatohepatitis, liver cirrhosis, advanced portal fibrosis, periportal necrosis, splenomegaly, hypertrophic cardiomyopathy, angina pectoris, a cardiac murmur
**Neurodegenerative phenotypes**	Intellectual impairment, delayed language ability, psychomotor retardation, moderate cognitive impairment, emotional excitability and hyperactivity, brisk patellar tendon reflexes, pyramidal signs, impaired motor skills, loss of ambulation, spastic gait, ataxic gait, waddling gait, intention myoclonus, dystonic tetraplegia with continuous myoclonus, generalised dystonia, progressive myoclonus epilepsy, seizures, died prematurely, peripheral neuropathy
**Changes in physical appearances**	Acanthosis nigricans, hirsutism, acromegaloid features, vitamin D resistant rickets, facial dysmorphism, odd shaped skull, “coarse” facial features, hollowed cheeks, peculiar pinched facies, triangular facies, abundant “kinky” scalp hair, premature greying of the hair, prominent forehead, synophrys, enophthalmos, big ears, bulbous nasal tip, wide mouth, macroglossia, prognathism, long fingers and toes, protruding abdomen, umbilical hernia, hernia, prominent calf muscles, high arched soles, rough dry skin, muscle hypertrophy, large superficial veins, contracture of joints, stiffness in joints, generalised eruptive xanthomas, mild anaemia
**Changes in reproductive system**	Precocious puberty, hypertrophic genitalia, enlarged penis, retractile testes, teratozoospermia, clitoromegaly, breast enlargement, oligomenorrhea, polycystic ovary, intrauterine growth restriction, polyhydramnios, persistent Mullerian ducts
**Changes in teeth and bone**	Severe crowding of maxillary and mandibular arches, labial exclusion of 33 and 43, aberrant crown morphology, caries, several teeth missing, generalised plaque induced gingivitis, bone cysts, advanced bone age in the first year of life, hyper-density of partial bones, skeletal abnormalities, diffuse osteosclerosis, well-defined osteolytic lesions sparing the axial skeleton, high serum sclerostin, good bone microarchitecture, high bone mineral density in trabecular sites, multiple bone lytic and pseudo-osteopoikilosis lesions limited to the hands and feet
**Others**	Developmental delay, multinodular goitre, failure to thrive, significant developmental delay, accelerated growth, chronic nasal congestion, enlarged tonsils and adenoids, voracious appetite, gum bleeding repeatedly, advanced bone age, pancreatitis, upper respiratory tract infection, elevated urinary organic acid, growth disorder, constipation, thrombocytopenia, cardiovascular autonomic neuropathy, papillary thyroid carcinoma

**Table 3 biomolecules-12-00840-t003:** Leptin-based treatment and pioglitazone treatment for *BSCL2*-mutation-related human diseases.

Studies	Disease	Patients	Treatment	Treatment Effects
Musso et al., 2005 [96]	CGL2	2	r-metHuLeptin (0.02–0.08 mg/kg/d, s.c., b.i.d, for 12 mo)	A 12-year-old female patient had amenorrhea prior to therapy and commenced normal menses after the therapy. The testosterone level in a male patient was not affected by this treatment.
Ebihara et al., 2007 [113]	CGL2	3	r-metHuLeptin (0.02–0.08 mg/kg/d, s.c., b.i.d, for 2, 8, and 24 mo, respectively)	Diabetes mellitus, insulin resistance, hypertriglyceridaemia, and fatty liver in CGL2 patients were ameliorated significantly by this treatment.
Beltrand et al., 2007 [114]	CGL2	6	r-metHuLeptin (0.015–0.086 mg/kg/d, s.c., q.d, for 4 mo)	In non-diabetic children with CGL2, insulin resistance, hypertriglyceridaemia, and fatty liver were significantly improved by the therapy.
Beltrand et al., 2010 [115]	CGL2	6	r-metHuLeptin (0.06–0.12 mg/kg/d, s.c., q.d, for 28 mo)	Though the treatment ameliorates metabolic disorders in some patients with CGL2, some patients may develop a resistance to leptin.
Araujo-Vilar et al., 2015 [116]	CGL2	7	Metreleptin (0.05–0.24 mg/kg/d, s.c., b.i.d or q.d, for 9–60 mo)	Metreleptin was effective for metabolic disorders caused by lipodystrophy including diabetes mellitus, hypertriglyceridaemia, and hepatic steatosis, without marked side effects.
Muniyappa et al., 2017 [117]	CGL2	6	Metreleptin (0.05–0.09 mg/kg/d, s.c., for 16–32 weeks)	Elevated plasma levels of ANGPTL3 in CGL2 patients was attenuated with leptin therapy.
Maeda et al., 2019 [118]	CGL2	2	Metreleptin (0.04–0.08 mg/kg/d, s.c., for 20 years), dietary control, medication, and psychosocial counselling	Metreleptin was the main line of treatment to ameliorate metabolic disorders in these two patients.
Araújo-Vilar et al., 2018 [119]	PELD	1	Metreleptin (0.03–0.08 mg/kg/d, s.c., b.i.d, for 54 mo) and high PUFA diet	The treatment significantly delayed the neurological regression and death of this patient.
Pedicelli et al., 2020 [120]	PELD	1	Metreleptin (0.06 mg/kg/d, s.c., for 2 mo)	The treatment not only treated metabolic disturbance but allowed better seizure control in this child.
Victoria et al., 2010 [111]	CGL2	1	Pioglitazone (4–8 mg/d, p.o., q.d, for 1 year)	Pioglitazone significantly improved glycaemic and lipid control, insulin sensitivity and serum leptin levels in this patient.
Chaves et al., 2021 [112]	CGL2	1	Pioglitazone (15 mg/d, p.o., b.i.d, for 5 years)	Lipodystrophy and insulin resistance were improved by it.

Abbreviations: CGL2: type 2 congenital generalised lipodystrophy, r-metHuLeptin: recombinant methionyl human leptin; s.c.: subcutaneous, b.i.d: bis in die; q.d: quaque die; Metreleptin: recombinant analogue of human leptin; ANGPTL3: angiopoietin-like protein 3; PELD: progressive encephalopathy with/without lipodystrophy; PUFA: polyunsaturated fatty acids; p.o.: peros.

**Table 4 biomolecules-12-00840-t004:** Metabolic disorders of seipin deficiency or overexpression in rodents.

Studies	Disease	Species (Gender)	Seipin Knockout	Method of Knockout	Treatment	Time of Assessment	In Vivo Findings
Cui et al., 2011 [121]	CGL2	Mice (M)	SKO	ESCs (HR)	/	12 weeks old	This was the first mouse model lacking seipin, which replicated most of the characteristics of patients with CGL2 including most adipose tissue loss, liver steatosis, glucose intolerance, and hyperinsulinaemia.
Chen et al., 2012 [47]	CGL2	Mice (F and M)	SKO	ESCs (HR)	/	6–13 weeks old	Seipin played an important role in lipolysis through the cAMP/PKA pathway in a cell-autonomous way.
Cui et al., 2012 [129]	ND	Mice (M)	hSeipin-aTg	Transgene	/	3–8 months old	Overexpression of seipin in adipocyte tissue reduced the fatty mass and the size of adipocytes and lipid droplets, and that might be related to elevated lipolysis.
Prieur et al., 2013 [122]	CGL2	Mice (M)	SKO	ESCs (HR)	Pioglitazone (45 mg/kg/d, p.o., for 9 weeks)	13 weeks old	Seipin played a key role in the differentiation and storage capacity of white adipocytes. Hypotriglyceridaemia was unique in SKO mice, which were linked to increased uptake of triglyceride-rich lipoprotein by the liver. Pioglitazone treatment partially rescued diabetes, insulin resistance, and adipokine levels in SKO mice.
Liu et al., 2014 [44]	CGL2	Mice (M)	aKO	ESCs (HR)	Rosiglitazone (0.3 mg/g in diet, for 10 weeks)	3–10 months old	Loss of seipin in adipocyte tissues resulted in insulin resistance and hepatic steatosis. Pioglitazone significantly alleviated metabolic disorders including insulin resistance in seipin-aKO mice.
Chen et al., 2014 [51]	CGL2	Mice (M)	SKO lKO	ESCs (HR)	Fasting (4 or 16 h)	11–13 weeks old	Liver lipid accumulation was not observed in seipin-lKO mice. Fasting for 16 h significantly sensitised hepatic insulin signalling in SKO mice.
Xu et al., 2015 [130]	CGL2	Mice (F and M)	SKO	ESCs (HR)	2% n-3 PUFAs (20 g/kg in diet, for 12 weeks)	20 weeks old	N-3 PUFAs therapy reduced triglyceride synthesis and enhanced β-oxidation in liver; thus, ameliorating hepatic steatosis and insulin resistance in SKO mice.
Gao et al., 2015 [131]	CGL2	Mice (F and M)	SKO	ESCs (HR)	Adipose-specific seipin reconstitution	12 weeks old	Most metabolic disorders in SKO mice were caused by seipin loss in adipocyte tissues. Dyslipidaemia, lipodystrophy, hepatic steatosis, and insulin resistance was nearly abolished by adipose-specific seipin reconstitution.
Zhou et al., 2015 [46]	CGL2	Mice (M)	maKO	ESCs (HR) with TAM induced maKO at 8–10 weeks old	/	20 weeks old	Acquired loss of seipin in adult mature adipocytes affected whole-body energy homeostasis through accelerating lipolysis and β-adrenergic signalling.
Dollet et al., 2016 [132]	CGL2	Mice (M)	SKO	ESCs (HR)	/	/	Seipin guaranteed the response to insulin and cold-activated adrenergic signals of brown adipose tissue through white adipose tissue.
Wang et al., 2016 [133]	CGL2	Mice (M)	SKO	ESCs (HR)	Ldlr KO; rosiglitazone (0.3 mg/g in diet, for 8 weeks)	20 or 28 weeks old	Ldlr KO induced new metabolic complications including hyperlipidaemia and atherosclerosis in SKO mice. Rosiglitazone treatment alleviated hyperlipidaemia and atherosclerosis in seipin and LDLR double knockout mice.
Dollet et al., 2016 [134]	CGL2	Mice (NA)	SKO	ESCs (HR)	FGF21 analogue LY2405319 (1 mg/kg/d, s.c., for 28 d)	10 weeks old	Seipin deficiency induced chronic activation of the p38 MAPK pathway in adipocytes. FGF21 administration elevated plasma adiponectin levels and normalised insulin sensitivity in SKO mice.
Joubert et al., 2017 [135]	CGL2	Mice (NA)	SKO	ESCs (HR)	SGLT2 inhibitor Dapagliflozin (1 mg/kg in water, for 8 weeks); Pioglitazone (45 mg/kg/d, p.o., for 8 weeks)	14 weeks old	Glucotoxicity could induce cardiac dysfunction by itself. In SKO mice, the O-GlcNAcylated protein levels in heart could be decreased by both dapagliflozin and pioglitazone, but only dapagliflozin successfully prevented the development of hypertrophic cardiomyopathy.
McIlroy et al., 2018 [136]	CGL2	Mice (M)	aKO	ESCs (HR)	/	12–16 weeks old	Loss of seipin in adipocytes did not result in severe metabolic disorders that were observed in CGL2 patients and SKO mice.
McIlroy et al., 2018 [137]	CGL2	Mice (F)	aKO	ESCs (HR)	/	6–28 weeks old	After 4 weeks of HFD feeding and 9 weeks thermoneutrality condition stay, seipin-aKO female mice showed a subtle alteration in metabolic homeostasis manifestation.
Liao et al., 2018 [138]	CGL2	Mice (F and M)	SKO	ESCs (HR)	apoE KO	9 months old	Seipin-deletion-induced metabolic disorders were independent of genetic background and experimental diet. Seipin deletion worsened apoE KO-induced atherogenesis.
Liu et al., 2018 [124]	CGL2	Mice (M)	SKO	ESCs (HR)	Adipose tissue transplantation at 3 months old; recombinant mouse leptin (1 μg/g/d, i.p., for 2 weeks)	6 months old	In SKO mice, glucolipotoxicity caused renal injury with impaired renal reabsorption. Metabolic disturbance and renal injury in SKO mice were significantly improved by adipose tissue transplantation and leptin administration.
Wang et al., 2019 [139]	CGL2	Mice (F)	SKO	ESCs (HR)	Adipose tissue transplantation at 6 weeks old	22 weeks old	Most metabolic disorders in SKO mice were caused by loss of adipose tissue. Adipose tissue transplantation reversed hepatic steatosis, insulin resistance, and dyslipidaemia in SKO mice.
Xu et al., 2019 [140]	CGL2	Mice (M)	SKO mKO	ESCs (HR)	/	10–15 weeks old	Muscle metabolic defects were observed in SKO mice but not in seipin-mKO mice. Muscle metabolic defects were caused by incomplete FAO in muscle, circulating insulin, and NEFA levels instead of seipin loss in muscle.
Zhou et al., 2019 [48]	CGL2	Mice (M)	SKO	ESCs (HR)	ATGL KO	10 weeks old	IGF1R-mediated PI3K/AKT pathway and elevated ATGL in hearts were related to heart disease of SKO mice. ATGL ablation significantly rescued lipodystrophy, insulin resistance and heart disease in SKO mice.
Bai et al., 2019 [123]	CGL2	Mice (M)	SKO cKO	ESCs (HR)	/	8–46 weeks old	Left ventricular concentric hypertrophy without hypertension occurred in SKO mice but not in seipin-cKO mice. Metabolic disorders (18–24 weeks old): diabetes mellitus, insulin resistance, and cardiomegaly were noted in SKO mice at 18–24 weeks old. Underlying mechanisms for heart phenotypes (23–36 weeks old): cardiac titin phosphorylation and reactive interstitial fibrosis associated with neutrophil extracellular traps induced left ventricular stiffness were found in SKO mice. More obvious heart phenotypes in elder mice (41–46 weeks old): higher left ventricular end-diastolic pressure and heart tissue weight to body weight ratio in SKO mice were observed.
McIlroy et al., 2020 [52]	CGL2	Mice (F and M)	aKO	ESCs (HR)	Hepatic seipin-KO at 8–12 weeks old	12–16 weeks old	Hepatic seipin ablation was unlikely to worsen dysfunction in seipin-aKO mice significantly.
Xiong et al., 2020 [13]	CGL2	Mice (F and M)	SKO Seipin^+/−^	ESCs (HR)	Rosiglitazone (5 mg/kg/d, p.o., for 2 weeks) in Seipin^+/−^ male mice; oestradiol (5 μg/kg/d, s.c., for 2 weeks) in Seipin^+/−^ female mice after ovariectomy	12 weeks old	Seipin^+/−^ female mice after ovariectomy and seipin^+/−^ male mice showed glucose intolerance and deficits in insulin synthesis and secretion without adipose tissue loss or insulin resistance. Both rosiglitazone and oestrogen relieved the reduction of PPARγ expression in Seipin^+/−^ mice.
Gao et al., 2020 [127]	CGL2	Mice (M)	SKO	CRISPR/Cas9 system, at birth	GPAT3 KO	12 weeks old	Seipin deficiency increased the activity of GPAT3. Insulin resistance and hepatic steatosis in SKO mice were significantly ameliorated by GPAT3 KO.
Bruder-Nascimento et al., 2021 [141]	CGL2	Mice (M)	SKO	ESCs (HR)	Leptin (0.3 mg/kg/d, s.c., for 7 d) via an osmotic minipump	10–13 weeks old	SKO induced hypertrophic vascular remodelling and adrenergic hypercontractility in mice. Leptin treatment improved adrenergic hypercontractility in SKO mice.
Wang et al., 2021 [142]	CGL2	Mice (M)	SKO	ESCs (HR)	/	6 months old	Both contractility and relaxation of vascular were impaired in SKO mice, and that were related to the reduction in perivascular adipose tissue and adipose-derived relaxing factors, which was caused by increasing macrophage infiltration and ERS.
McGrath et al., 2021 [143]	CGL2	Mice (M)	SKO	ESCs (HR)	Exercising, for 6 weeks	18 weeks old	Seipin deficiency induced lower bone marrow adipose tissue in mice while exercising increased trabecular bone.
Zhou et al., 2022 [144]	CGL2	Mice (M)	cKO	ESCs (HR)	FAO inhibitor trimetazidine (15 mg/kg/d, i.p., for 6 weeks); High-fat diet for 3 mo; abolishment of one ATGL allele	24–30 weeks old	Systolic dysfunction with dilation developed in seipin-cKO mice, and that is related to increased ATGL expression and FAO and drastic reduction in cardiac lipidome. Trimetazidine, high-fat diet, or ATGL inhibition ameliorate cardiac dysfunction partially by inhibiting FAO or supplying lipid.
Ebihara et al., 2022 [145]	CGL2	Rats (M)	SKO	ENU mutagenesis	Pioglitazone (2.5 mg/kg, p.o., for 4 weeks); rosiglitazone (3 mg/kg, p.o., for 4 weeks)	16 weeks old	Neither pioglitazone nor rosiglitazone rescued diabetes mellitus, hypertriglyceridaemia, and fatty liver in SKO mice.

Abbreviations: CGL2: type 2 congenital generalised lipodystrophy; M: male; SKO: seipin knockout; ESCs: embryonic stem cells; HR: homologous recombination; F: female; cAMP: cyclic AMP; PKA: protein kinase A; ND: no data; hSeipin-aTg mice: transgenic mice overexpressing human Seipin in adipocyte tissue; p.o.: peros; aKO: adipocyte-specific knockout; lKO: liver-specific knockout; PUFAs: polyunsaturated fatty acids; maKO: mature adipocyte-specific knockout; TAM: tamoxifen; ENU: N-ethyl-N-nitrosourea; Ldlr KO: low-density lipoprotein receptor knockout; FGF21: fibroblast growth factor 21; s.c.: subcutaneous; MAPK: mitogen-activated protein kinase, SGLT2: sodium-glucose cotransporter 2, HFD: high-fat diet, apoE: apolipoprotein E, i.p.: intraperitoneal, mKO: muscle specific knockout; FAO: fatty acid oxidation; NEFA: non-esterified fatty acids; ATGL: adipose triglyceride lipase; IGF1R: type 1 insulin-like growth factor receptor; PI3K: phosphoinositide 3-kinase; AKT: protein kinase B; cKO: cardiomyocyte-specific knockout; PPARγ: peroxisome proliferator activated receptor gamma; GPAT3: glycerol-3-phosphate acyltransferase 3; ERS: endoplasmic reticulum stress.

**Table 5 biomolecules-12-00840-t005:** Neural disorders of seipin deficiency or overexpression in rodents.

Studies	Disease	Species (Gender)	Seipin Knockout	Method of Knockout	Treatment	Time of Assessment	In Vivo Findings
Zhou et al., 2014 [8]	CGL2	Mice (F and M)	nKO SKO	ESCs (HR)	17b-estradiol (120 μg/kg, s.c., for 20 d); rosiglitazone (5 mg/kg, p.o., for 30 d)	8–12 weeks old	Affective disorders occurred in seipin-nKO male mice and SKO mice via reducing PPARγ levels. Both 17b-estradiol and rosiglitazone alleviated affective disorders by rescuing PPAR_γ_ expression in male seipin-nKO mice.
Li et al., 2015 [125]	CGL2	Mice (F and M)	nKO	ESCs (HR)	Rosiglitazone (5 mg/kg/d, p.o., for 10 d); MEK inhibitor U0126 (0.3 nmoL/mouse/d, i.c.v., for 10 d)	12–16 weeks old	Neuronal seipin deficiency impaired proliferation and differentiation of neural stem and progenitor cells of hippocampal dentate gyrus by reducing PPARγ. Rosiglitazone significantly reduced the severity of depression-like phenotype in seipin-nKO mice through MAPK and Wnt3 pathways, and this alleviation was blocked by U0126.
Ebihara et al., 2015 [128]	CGL2	Rats (M)	SKO	ENU mutagenesis	/	/	Seipin was necessary for normal brain development.
Zhou et al., 2016 [9]	CGL2	Mice (M)	nKO aKO SKO	ESCs (HR)	Rosiglitazone (5 mg/kg, p.o., for 12 days) MEK inhibitor U0126 (0.3 nmol/3 μL/mouse, i.c.v., for 12 days) Trk family inhibitor K252a (0.2 μmol/3 μL/mouse, i.c.v., for 12 days)	12–14 weeks old	LTP and spatial cognitive deficits induced by activation of the ERK-CERB-AMPAR pathway were observed in SKO mice and seipin-nKO mice but not seipin-aKO mice. The activation of PPARγ by rosiglitazone treatment could rescue the hippocampal LTP and spatial cognitive deficits in SKO mice and seipin-nKO mice, and this alleviation was blocked by U0126 but not K252a.
Wang et al., 2018 [10]	CGL2	Mice (M)	nKO aKO SKO	ESCs (HR)	Rosiglitazone (5 mg/kg, p.o., for 28 d) GSK3β inhibitor AR-A014418 (1 mg/kg, i.p., for 28 days)	3–12 months old	The age-related deficit in motor coordination induced by dopaminergic neuron injury, which is caused by enhanced aggregation and phosphorylation of α-synuclein or neuroinflammation was found in seipin-nKO mice and SKO mice but not seipin-aKO mice. Rosiglitazone treatment greatly improved motor coordination by reducing α-synuclein oligomers, phosphorylation ofα-synuclein, and cleaved caspase-3; AR-A014418 only partially ameliorates motor coordination by attenuating the phosphorylation of α-synuclein.
Chang et al., 2019 [148]	CGL2	Mice (M)	nKO aKO SKO	ESCs (HR)	Rosiglitazone (4 mg/kg, p.o., for 7 days or 28 days); GSK3β inhibitor AR-A014418 (1 mg/kg, i.p., for 7 d); mTOR inhibitor rapamycin (1 μg/kg, i.p., for 7 d); PI3K inhibitor LY294002 (0.3 nmol/3 μL/mouse, i.c.v., for 7 d); JNK inhibitor SP600125 (10 mg/kg, i.c.v., for 7 d);	20–24 weeks old	The phosphorylation and aggregation of tau protein in seipin-nKO mice and SKO mice were more severe than that in seipin-aKO mice. Phosphorylation and aggregation of tau protein at Thr^212^/Ser^214^ and Ser^202^/Thr^205^ were abolished through rosiglitazone or A014418 or rapamycin or LY294002 in seipin-nKO or SKO mice, and hyperphosphorylated tau at Ser^396^ in SKO mice was corrected by rosiglitazone or SP600125.

Abbreviations: CGL2: type 2 congenital generalised lipodystrophy; F: female; M: male; nKO: neuron-specific knockout; SKO: seipin knockout; ESCs: embryonic stem cells; HR: homologous recombination; s.c.: subcutaneous; p.o.: peros; PPARγ: peroxisome proliferator activated receptor gamma; i.c.v.: intracerebroventricular; MAPK: mitogen-activated protein kinase; ENU: N-ethyl-N-nitrosourea; LTP: long-term potentiation; ERK: extracellular signal-regulated kinase; CERB: cyclic AMP response element-binding protein; AMPAR: α-amino-3-hydroxy-5-methyl-4-isoxazole propionic acid receptor; i.p.: intraperitoneal; GSK3β: glycogen synthase kinase-3β; mTOR: mammalian target of rapamycin.

**Table 6 biomolecules-12-00840-t006:** Reproductive system disorders of seipin deficiency or overexpression in rodents.

Studies	Disease	Species (Gender)	Seipin Knockout	Method of Knockout	Treatment	Time of Assessment	In Vivo Findings
Jiang et al., 2014 [14]	CGL2	Mice (M)	gKO aKO SKO	ESCs (HR)	/	/	Impaired phospholipid homeostasis and male infertility were observed in SKO and seipin-gKO mice but not in seipin-aKO mice.
El Zowalaty et al., 2015 [15]	CGL2	Mice (M)	SKO	ESCs (HR)	/	2–3 months old	Male infertility that was indued by Seipin deficiency was associated with increased spermatid apoptosis, increased chromocenter fragmentation, defective chromatin condensation, abnormal acrosome formation, and defective mitochondrial activity.
Li et al., 2015 [11]	CGL2	Mice (F)	SKO	ESCs (HR)	500 ppm genistein diet, for 2 weeks	5 weeks old	SKO female mice accelerated postnatal mammary ductal development but delayed vaginal opening. Genistein treatment accelerated vaginal opening but increased mammary gland area in SKO female mice.
Ebihara et al., 2015 [128]	CGL2	Rats (M)	SKO	ENU mutagenesis	/	/	Seipin was necessary for normal spermatogenesis.
El Zowalaty et al., 2017 [126]	CGL2	Mice (F)	SKO	ESCs (HR)	/	2–10 months old	Seipin deficiency in mice led to myometrial hypertrophy and defective parturition.
El Zowalaty et al., 2018 [12]	CGL2	Mice (F)	SKO	ESCs (HR)	/	2–4 months old	Seipin deficiency induced insufficient milk production during lactation, which is associated with increased ERS and apoptosis in mammary gland alveolar epithelial cells.

Abbreviations: CGL2: type 2 congenital generalised lipodystrophy; M: male; gKO: germ cells-specific knockout; aKO: adipocyte-specific knockout; SKO: seipin knockout; ESCs: embryonic stem cells; HR: homologous recombination; F: female; ENU: N-ethyl-N-nitrosourea; ERS: endoplasmic reticulum stress.

**Table 7 biomolecules-12-00840-t007:** The role of seipin in rodent disease models.

Studies	Disease	Species (Gender)	Seipin Genetic Status	Method	Treatment	In Vivo Findings
Zhang et al., 2017 [166]	Parkinson’s disease	Rats (M)	WT	6-OHDA (8 μg/4 mL in 0.9% saline containing 0.1% ascorbic acid) injected into the right substantia nigra pars compacta and ventral tegmental area	Echinacoside (3.5 and 7 mg/kg, i.p., for 14 d)	Seipin aggregation and ER stress were observed in the Parkinson’s disease model. Echinacoside significantly reduced 6-OHDA-induced nigrostriatal dopaminergic neuron ER stress by promoting seipin degradation.
Li et al., 2019 [16]	Hepatic steatosis	Mice (M)	WT	HFD fed	AAV9-*SEIPIN*	The effects of seipin on triglycerides and PGC-1α were dependent on calcium concentrations. AAV-mediated seipin overexpression alleviated HFD-induced hepatic steatosis.
Ren et al., 2020 [18]	Stroke (IRI)	Rats (M)	WT	IRI was established 1 h after McAO	MiR-187-3p * antagomir (1.5 nmol/mouse, i.c.v., 2 h before McAO)	IRI-caused cerebral damage was associated with elevated miR-187-3p expression and thus decreased seipin expression. MiR-187-3p inhibitor reduced infarction via upregulating seipin-mediated autophagy in the stroke model.
Ren et al., 2021 [19]	Stroke (IRI)	Rats (M)	WT	IRI was established 1 h after McAO	MiR-187-3p * antagomir (1.5 nmol/mouse, i.c.v., 2 h before McAO)	IRI induced cerebral damage was associated with seipin-mediated ER stress. MiR-187-3p inhibitor ameliorated IRI-induced cerebral injury by regulating seipin-mediated ER stress.
Qian et al., 2016 [20]	Neuroinflammation	Mice (M)	SKO	Aβ_25–35_ (1.2 mol/mouse, i.c.v.) or Aβ_1–42_ (0.1 nmol/mouse, i.c.v.)	Rosiglitazone (5 mg/kg/d, p.o., for 17 days)	Seipin deficiency in astrocytes worsened Aβ_25–35/1–42_-induced neuroinflammation and cognitive impairment via reducing PPARγ to increase GSK3β activity. Rosiglitazone treatment protected SKO mice from Aβ_25–35/1–42_-induced neuroinflammation and neurotoxicity.
Chen et al., 2016 [170]	Stroke (IRI)	Mice (M)	SKO	IRI was established 1 h after McAO	/	Seipin deficiency worsened IRI-induced cerebral injury via increasing BBB permeability, amplifying ER stress, increasing glucose levels, and decreasing adipokines levels.
Bruder-Nascimento, et al., 2019 [146]	CGL2	Mice (M)	SKO	/	Leptin (0.3 mg/kg/d, s.c., for 7 d)	Leptin-PPARγ-Nox1 pathway was involved in the endothelial dysfunction in SKO mice. Leptin replacement therapy fully restores impaired endothelium-dependent relaxation in SKO mice.
Wu et al., 2021 [21]	Heart failure	Mice (M)	SKO	Transverse aortic constriction for 12 weeks	/	Seipin deficiency worsened cardiac hypertrophy and diastolic heart failure which may be related to the impairment of myocardial calcium handling, ER stress, inflammation, and apoptosis in the heart failure model.

* MiR-187-3p causes post-transcriptional gene silencing by binding to the protein-coding sequence of seipin. Abbreviations: M: male; WT: wild type; 6-OHDA: 6-hydroxydopamine; i.p.: intraperitoneal; ER: endoplasmic reticulum; HFD: high-fat diet; AAV: adeno-associated virus; PGC-1α: peroxisome proliferator-activated receptor-γ coactivator-1α; IRI: ischaemia/reperfusion injury; McAO: middle cerebral artery occlusion; i.c.v.: intracerebroventricular; SKO: systemic knockout; Aβ: β-amyloid; p.o.: peros; PPARγ: peroxisome proliferator activated receptor gamma; GSK3β: glycogen synthase kinase-3β; BBB: blood–brain barrier; CGL2: type 2 congenital generalised lipodystrophy; s.c.: subcutaneous; Nox1: NADPH Oxidase 1.

## Data Availability

Not applicable.

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
