# Peer review of "Role of Seipin in Human Diseases and Experimental Animal Models"

_biomolecules, 2022, doi:10.3390/biom12060840_

Round 1

Reviewer 1 Report

Review of manuscript of Seipin paper (1741998) for Biomolecules (May 2022)

This paper is a review of the BSCL2 gene product Seipin, the diseases caused by mutations in the encoding BSCL2 gene (in humans and in animal models), and their possible treatment options.  There are several points for attention:

1. It is a comprehensive and up-to-date review which will be helpful, though does tend to make rather heavy reading in some places from being effectively a list of published observations, which can tend to detract from the thread of underlying narrative.  The authors should consider whether some sections might be helped by addition of a table summarising the observations (eg. in section 3.1 on CGL2).  

2. Also, the written English requires multiple corrections throughout, currently adding to the difficulty of reading the article. 

3. Scientifically, the review seems accurate and sound, although in section 4.2 (‘Role of seipin in Seipinopathies’) the authors need to make very clear whether the mention of a single variant (eg. N88S) is indicating that this is present heterozygously or homozygously. If it is as a homozygote, it should be written consistently as N88S/N88S (or ‘homozygous N88S’).  Thus N88S/S90L is presumably a compound heterozygote, but is ‘N88S and S90L’ in line 402 referring to each variant as an isolated heterozygote, or each as homozygous, or both together as a compound heterozygote ?   These need to be clarified throughout the whole of section 4.2.  Also: The heading for section 4.2 needs to be changed to:  ‘Role of different BSCL2 variants in Seipinopathies’, rather than the having the tautology of the present section title.

4. Lines 198-211 ‘Seipinopathies’ need similar consideration. Eg. line 198 : ‘Heterozygous missense mutations (N88S/S90L) in BSCL2 cause a broad spectrum of motor neuron diseases, not CGL2’.  Do the authors mean when these mutations are singly present as heterozygotes , or do they mean N88S/S90L as a compound heterozygous combination ?  Each of the mutations in line 209 is said to cause a single patient with a disease of the motor neurone.  Are these as heterozygous mutation, and if so why have they been picked out from all the many other heterozygous BSCL2 mutations that can be associated with a motor neuropathy phenotype ?  The one of most interest is perhaps N88T if this can cause a motor neuropathy as a heterozygote whereas N88S causes CGL2 when homozygous.  If that is correct, the authors should comment further on this.  

5. This is of importance as it is evident from Table 1 that CGL2 phenotypes are almost all inherited as autosomal recessive, whereas the neuropathy phenotypes are almost all Autosomal Dominant. This needs to be made very clear and early-on in the article – perhaps in the 2nd paragraph of the Introduction, possibly under a sub-heading of ‘Genetic inheritance basis’.

6. Section 6 ‘Lessons learned from experimental animals’.  The 2nd paragraph of this section describes 5 different modalities for restoring adipose tissue function.  To avoid a tendency to lose one’s way in this paragraph, it would help to start the paragraph by listing the 5 modalities, then amplifying in the text with a description of each in turn.  Eg. ‘Line 474-5:  We summarise the 6 potential modalities for restoring adipose tissue function:  1) (heading only); 2) (heading only); 3 (heading only)…  .  Details are as follows: 1)…..

7. Line 531 is the first mention of other proteins involved in the endoplasmic reticulum which can be involved in other related diseases. It would help perspective if the Introduction could say that Seipin is one of a number of proteins found in the ER that are involved in LD biogenesis.

8. Line 218-9. Leptin-replacement therapy is said to ‘develop’ normal menses in a 12yr-old girl with CGL2 and amenhorrhea. Since this girl was aged only 12 years, how did the original authors (ref 89) know this was causative rather than coincidental ? A few brief words on the evidence for that is required if this is to be included here.  If the evidence is weak, it would be better to say ‘was associated with’ rather than ‘develops’.  If the evidence is proven, I would suggest using ‘induces’ rather than ‘develops’.

9. Figure 1 and Figure 2.  Are these new Figures original to this article ?.  If not, the source must be appropriately acknowledged and referenced.

10. Supplementary materials.  The given download address: www.mdpi.com/xxx/s1 does not work, but produces a 404 error message.  It is not possible therefore to review the supplementary material.   

More Minor corrections required to English language

11. Unfortunately these are very numerous, and over 60 from the main text (excluding any in the authors’ Tables) are listed in the Table below. This is also attached as a separate Word file.  After attending to these, the paper should still be proof-read/corrected for the English.

Biomolecules 1741998 Seipin English correction Table

Line

Written

Change to (or comment)

27

 upper- and down-stream

upstream and downstream

59

While remains to be elusive,

A word is missing here. Do the authors mean – ‘While the complete function of seipin remains elusive, existing studies have outlined its predominant function as follows:…’   ?

73

structuring

structural

79

play

plays

92

interact

interacts

107

Please differentiate the heading of Figure 1 (and Figure 2) by bold type or underlining.

107

Organ damages

Organ damage  or   organs damaged

108

product

produce

108

induce

induces

110

resulted

resulting

110

neurons including

Add comma:   ‘neurons, including’

111

impairment also

impairment, are also

113

atrophy distal muscle groups

atrophy of distal muscle groups

138

shift

frameshift

140

but few

but a few

141

are

being

149-150

higher risk of premature death [51,52,70].

? ….than with other forms of CGL ?

150

peroxidation are

peroxidation is

157

higher morbidity

higher morbidity than whom ?

162

but it may due to

but these may be due to

162-163

only one each case has

Only one case of each problem has

167

arrow

Typo:  ? ‘marrow’

205

there are

there is

206

exists

observed

214

reported

which report

218

develops

was associated with   (or  induced ?).  See main comment 8 above.

250-251

may due to

may be due to

252

In accord

Delete ‘In accord’.  Start sentence with ‘Cardiomyopathy’

267

blood lipids metabolic disorders

blood lipid patterns in metabolic disorders

271

its

the

273

its

the

274

This review only complements some other

this review complements this by adding some other

276

cause

causes

277

via

by

279-280

On one hand, there is following evidence support the conclusion

On the one hand, the following evidence supports the conclusion

290-291

failed to cause

failed to show

292

they developed

did they develop

293

put

puts

309

vessels contractility

vessel contractility

322

studies focused

studies have focused

374

resultant

and resultant

376

phenotypes of

phenotypes involving the

377

includes

include

380-381

it may have an anti-proliferative function as it is in the uterus

Do the authors mean:  ‘it may have an anti-

proliferative function as it does in the uterus’  …?  If so, please change ‘is’

382

are responsible for

This implies they are the only factors responsible.  Better to put: ‘have a role in’

383-384

seipin involved mature LDs formation

seipin-involved formation of mature LDs

385

protect

protects

389

…should be responsible for it [124]

could both be expected to be responsible for this [124].

401

is in debating

‘is a subject of debate’  or  ‘is a matter of debate’

404

results

resulting

408

result in inclusion bodies formation

results in the formation of inclusion bodies

410

at 25 to 30 percent

What does this mean ?  Is it in 25-30% of motor neurons , or at a 25-30% level compared with a normal cell. Please specify.

415-416

different with

different from

416-417

because it either recruit components which is necessary for aggresomes nor assemble at microtubule organising centres

This sentence is incomprehensible.  Please rewrite.

420

lead to

leads to

442

suppress

suppresses

455

size and that

Delete: ‘and’  :  to read ‘size that’

465

is in paucity

Either write:  ‘is lacking’ (if there is no research), or write ‘is very limited’

488-489

In parallel in leptin, another

In parallel, leptin, another

490

plasm

Typo:  plasma

496

reverse

reverses

499

which

where

522

prospective

Do the authors mean ‘perspective’ ?

533

so on also

so on are also

538

focus

‘focused’  or ‘focusing’

Reviewer 2 Report

Yuying Li et al. present an extensive and complete review of the diseases related to pathogenic variants in the BSCL2 gene, including the main clinical manifestations, their pathogenetic mechanisms and potential therapeutic approaches, as well as an extensive summary of the Bscl2 transgenic animal models generated to date.

This is a relevant and very interesting review where, in general, both the clinical aspects and the molecular bases of BSCL2-associated diseases are analyzed in sufficient depth.

Minor comments

  1. Although the term “seipinopathies”, as the authors comment, was coined by Ito to refer to diseases of the first and second neuron associated with heterozygous missense variants in the BSCL2 gene, it is not actually correct. In my opinion, all disorders that are a consequence of pathogenic variants in BSCL2-seipin should be called seipinopathies, in the same way that diseases caused by pathogenic variants in the LMNA gene, which encodes the Lamin A/C protein, are called “laminopathies”. From my point of view, it makes no sense to reduce the term seipinopathy to only one type of disorder, which could well be called BSCL2-associated motor neuron diseases, according to a dyadic nomenclature.

  2. Although the description of the functions of seipin is well developed and touches on its fundamental aspects, some comment on the tissue-specific functions of seipin is missing (for example, adipose tissue vs. liver in relation to neutral lipid storage function);  or the functions of seipin in the central nervous system, beyond the findings in transgenic mice -although in relation to this, the authors make a detailed review of the mechanisms related to neurodegeneration in section 4.1.2.-). With current knowledge, it seems clear that seipin plays an important role in the synthesis of phospholipids, in this sense, the authors do not believe that defects in this function could alter the formation of synaptic vesicles, and their priming machinery necessary for the release of neurotransmitters, thus contributing to the neurological phenotype? Neither there is a mention to the role of seipin (Wang S et al. Nat. Commun. 2018, 9, 2939) in the peroxisomes biogenesis and its putative relation with neuron damage.

  3. The authors do not mention the cases of epileptic encephalopathy associated with variants in BSCL2 in heterozygosity, of which several cases have been reported ( Fernandez-Marmiesse, A et al. Seizure 2019, 71:161–165, Stanley NE et al. J Neuropathol Exp Neurol. 2022;81:377-380).

  4. It would be interesting to include a paragraph regarding the differential diagnosis with other subtypes of CGL (GPAT2, ,CAV1, PTRF).

  5. Massive hepatomegaly associated with steatosis, along with normal triglycerides, are common findings in SKO mice. However, there is no evidence that these animals end up developing cirrhosis (and rarely fibrosis), unlike what happens in humans with CGL2. Beyond the inter-species differences, these findings do not cease to surprise. Dare the authors speculate why this happens? The liver would be acting as a “buffer”, storing neutral lipids, which would explain why triglycerides are normal or even low (as they rightly comment in their manuscript), and that in mice the capacity to store lipids is much higher than in humans. What would justify that there was no evolution to cirrhosis?

Round 2

Reviewer 1 Report

Comments on revision of Biomolecules 1741998 (09.06.22)

This revised and resubmitted paper now provides a comprehensive review of the role of seipin in human metabolic and neurological diseases, together with animal models and treatment options. It is much improved from the original, particularly in the English language, and with having the extra Tables.  One or two aspects still require minor corrections.  These are:

1. Line 125: ‘defective synaptic vesicle budding’ rather than ‘defect…….’

2. Table 1.  Several of these entries have no OMIM number, and/or have an inheritance pattern marked as NA (not available), yet are listed as having missense or nonsense mutations, or as having a specific diagnosis (eg. 9 patients with PELD). If the specific underlying gene is known, there will be an OMIM number to go with cases that have mutations in that gene.  Similarly, if it cannot be determined whether the inheritance is AD or AR, and either could occur in that diagnosis, it would be better to indicate that specifically (eg. as ‘Either AR or AD’), rather than as ‘not available’.

3.  Line 191: ‘poorer prognosis’ rather than ‘poor prognosis’.

4. Table 2, 1st line:  typo:  ‘hyperlipidaemia’

5. Line 230:  a special nonsense mutation-c.985C>T mutation’.   No need to repeat ‘mutation’ - delete the 2nd ‘mutation’.  It is also important to say here that the mutation can be homozygous or in compound heterozygosity with other mutations (see ref 69).

6. Line 240: ‘…according to the rule of name diseases…’   It would be better to write here:  ‘…following the terminology used for diseases…'

7. Lines 264-266.  There is a problem with this. A 12-year old cannot be said to have 'primary amenorrhea' when the non-starting of her menses by age 12 years could be purely physiological. The authors should re-write this sentence as (perhaps) : 'A 12-year old female CGL2 patient who had not yet started her menses, did do so after receiving leptin.'

However, in the paper of reference 96 (Musso et al, 2005)(their Table 1), the 12yr-old is the only female CGL patient described in that Table who is known to have a BSCL2 mutation.  The other 6 females with CGL have AGPAT2 mutations, and so would be CGL1. Since they are mentioned in the present text under the heading of:  3.4. Treatment for human diseases caused by BSCL2 mutations’, the authors need to make very clear that their ‘…seven female patients (15-40 years old) with lipodystrophy…’ are in fact 6 patients with CGL1 associated with AGPAT2 gene mutation.

8. Line 266.  ‘…has induced regular menses…’  would be better written as ‘…have commenced regular menses…’  

 9. Table 3.  Similarly for the 1st patient entry (Musso et al):  ‘…and commenced normal menses…’  rather than  ‘…has induced normal menses’
